# Utilization of Lumpfish (*Cyclopterus lumpus*) Skin as a Source for Gelatine Extraction Using Acid Hydrolysis

**DOI:** 10.3390/md22040169

**Published:** 2024-04-10

**Authors:** Abhilash Sasidharan, Elise Rabben Tronstad, Turid Rustad

**Affiliations:** 1Department of Biotechnology and Food Science, Norwegian University of Science and Technology (NTNU), 7491 Trondheim, Norway; abhilash.sasidharan@ntnu.no (A.S.); elise_tronstad@hotmail.com (E.R.T.); 2Department of Fish Processing Technology, KUFOS, Kochi 682506, India

**Keywords:** lumpfish, collagen, gelatine, acid hydrolysis, gel strength, viscosity

## Abstract

Lumpfish (*Cyclopterus lumpus*) is an underutilized marine resource that is currently only being exploited for roe. Lumpfish skin was pre-treated with alkali (0.1M NaOH) and acid (0.1M HCl) at a skin to chemical ratio of 1:10 for 24 h at 5 °C to remove non-collagenous proteins and minerals. The pre-treated skin was washed, and gelatine was extracted with 0.1M of acetic acid at three different ratios (1:5, 1:10, and 1:15), time (12,18, and 24 h), and temperature combinations (12, 28, and 24 °C). The highest total extraction yield (>40%) was obtained with combinations of extraction ratios of 1:15 and 1:10 with a longer time (24 h) and higher temperature (18–24 °C). The highest gelatine content was obtained with an extraction period of 24 h and ratio of 1:10 (>80%). SDS-PAGE analysis confirmed the presence of type-I collagen. A rheological evaluation indicated melting and gelling temperatures, gel strength, and viscosity properties comparable to existing cold-water gelatine sources.

## 1. Introduction

Lumpfish (*Cyclopterus lupus*) is a marine cold-water fish that belongs to the genus Cyclopterus from the family Cyclopteridae, with a major distribution in the North Atlantic to the Barents Sea, around Iceland, Norway, Greenland, and Canada [1,2]. It is a commercially important species, valued for its roe and its application as a cleaner fish in salmon farming [2]. Wild-caught lumpfish are harvested primarily for their roe, as it is a delicacy in European markets. The rest of the carcass is generally considered as the residual raw material due to the absence of any consumer preference [1]. Except in Iceland and partly Greenland, wild-caught lumpfish carcasses are generally discarded in the sea after roe harvesting [1]. Iceland, as an exception, has started to catch wild lumpfish and export them frozen to the Asian food market [3]. However, the results from a test market study of lumpfish as a food item conducted in South Korea show the task to be challenging due to the peculiar appearance, taste, texture, and feeding preferences of the fish [4]. Lumpfish is used as a cleaner fish (to remove lice) in aquaculture, and in 2020, around 35 million lumpfish were used in Norway. When salmon is slaughtered, the lumpfish is 4–500 g and has stopped eating lice. The lumpfish is wasted, destroyed, or used to produce silage [5].

The European Waste Framework Directive 2008 and recent amendments (EU WFD (2018/851)) [6] stipulate comprehensive regulations that the manufacturers/fishermen must follow while handling the side streams generated during processing and capture. This demands a detailed compilation of information regarding the yield, classification, and valorization potential of side streams associated with the explored cold-water finfish species. However, this has opened new opportunities for the industry to further explore the potential for side-stream valorization. In 2020, around 42% of marine side streams, including lumpfish carcasses generated in Norway, was utilized for silage production, which is subsequently used as fish feed, livestock feed, and biogas/energy [5].

Collagen is the major structural protein found in large concentrations in seafood side streams, such as the skin, bone, and scales. The skin remains the most important source for marine collagen extraction, even though it can be isolated from different side streams, such as the scales, bone, head, air bladder, and viscera [7]. In lumpfish, the skin and head contribute up to 40–50% of the lumpfish carcass weight after roe extraction. The presence of amino acids, such as glycine, proline, alanine, and glutamic acid, indicates that lumpfish side streams can be a potential source of marine collagen [3]. Up to 29 types of collagen are identified from different types of tissue. Type I, which is extracted from raw materials, such as animal hide and fish skin, finds the highest application in the biomedical and cosmetic industries, owing to its lower antigenic and low cell-compatible properties [7]. Even if bovine sources are still the main source for extracting type-I collagen [8], different health, cultural, religious, and social preferences have led to the exploration of alternative collagen raw material resources, such as fish skin [9]. Irrespective of the advantages, marine collagen has some challenges that need to be addressed. Properties like a low denaturation temperature, connected to a low proline and hydroxyproline content, etc., limit its application when compared to bovine collagen [10]. Collagen from cold-water fish also has lower melting and gelling temperatures than warm-water fish collagen [9]. Conventional extraction methods depend on the solubility properties of the collagen and consist of salt-, acid-, or enzyme-aided processes [11]. Parameters such as the extraction temperature, acid concentration, extraction period, and skin to acid ratio have been found to influence the yield and gelatine quality [12]. This study focuses on the acid-aided process using acetic acid to extract gelatine from lumpfish skin aiming to propose an optimum process considering the different extraction combinations for the maximum yield and quality of gelatine.

## 2. Results and Discussion

### 2.1. Yield and Proximate Composition of Lumpfish Carcass

The lumpfish carcass consisted of 41.4% skin, 34.6% bone, and 24.0% muscle tissue (Table 1). This indicates that the major portion of the lumpfish carcass is skin, making it a suitable raw material for collagen extraction. The yield data agrees with the observations made by Ageeva et al. [3]. The proximate composition of fish is affected by different factors, such as size, feeding habits, season, and sexual maturity [3]. The proximate composition indicated the presence of a high level of moisture in all the three fractions (82–86%), which was also observed by other researchers [3,13]. The skin had the highest amount of protein (11.66 ± 0.65%) and the lowest amount of fat (0.62 ± 0.11%). The meat, on the other hand, presented the highest amount of fat (10.72 ± 0.35%). This was previously observed by Davenport and Kjørsvik [14], as they pointed out that lumpfish obtains neutral buoyancy in water, despite the absence of a swim bladder, by maintaining a high water content and fat deposition in the muscle. The higher ash content in the skin and frames also agrees with other studies [3], and indicates the need for a de-mineralization process during the pre-treatment stage.

### 2.2. Amino Acid Profile and Hydroxyproline Content of Lumpfish Skin

Glycine is considered as the dominant amino acid in fish skin, and the results (Table 2) also agree with this observation. Ageeva et al. [3]. Conducted the amino acid profiling of whole lumpfish and made similar observations regarding the domination of glycine in the amino acid profile. Other researchers also observed that the most abundant amino acid in fish skin is glycine [15,16]. Hydroxyproline is an amino acid found almost exclusively in collagen. When the content of hydroxyproline in pure collagen from the relevant species is known, the hydroxyproline content can be used to determine the collagen content, and the amount of hydroxyproline in a collagen or gelatine sample can be used to determine its purity [17]. The high content of glycine together with a high content of hydroxyproline indicates that the lumpfish skin can be utilized as a source for extracting collagen and gelatine [18]. The hydroxyproline content is also important for the physical properties of both collagen and gelatine [19]. The hydroxyproline content in freeze-dried lumpfish skin was found to be 4 + 0.09%. Fish skin generally contains low concentrations of hydroxyproline compared to mammalian skin [11], especially in the case of cold-water fish species [20]. The result indicates that the hydroxyproline content in lumpfish skin used is lower than in the skin samples from Atlantic cod (*Gadus morhua*) (5.1%) [21] and Atlantic salmon (*Salmo salar*) (7.51%) [22], two of the common cold-water fishes from the region. 

### 2.3. Total Extraction Yield

Total extraction yield (in % of dry weight of the skin) is represented in Figure 1 as a function of time, duration, and ratio. The results show that a higher extraction time and extraction temperature result in a higher extraction yield. Sample G15-24-18 (ratio 1:15, 24 °C, and 18 h) (41.1%) and sample G10-24-24 (ratio 1:10, 24 °C, and 24 h) (38.1%) had the highest total extraction yield, while sample G5-12-18 (ratio 1:5, 12 °C, and 18 h) (7.3%) reported the lowest. Extraction time and temperature were observed to have a significant influence (*p* < 0.01) on the total extraction yield. Extraction for 24 h produced a significantly higher (*p* < 0.01) yield than the extraction for 12 h. Arnesen and Gildberg [23] reported similar extraction yields for cold-water fish salmon (39.7 ± 2.2%) and cod (44.8 ± 0.2%) using acid hydrolysis at higher temperatures (56 °C). It may also be relevant to extend the extraction time, but according to Boran and Regenstein [17], a longer extraction time usually results in a higher extraction yield and protein yield at the expense of purity, compared to a shorter extraction time. 

### 2.4. Total Amino Acid Composition of Isolated Gelatine

The total amino acid content of each sample is presented in Table 3. The total amino acid content is a measure of the protein content [24]. The highest total amino acid content was found in G15-12-12 with 946 ± 141 mg/g and the lowest total amino acid content was G5-12-18 with 615 ± 156 mg/g. The results show that glycine/arginine and tyrosine are the amino acids that form the largest proportion in the freeze-dried samples. Sample G5-12-24 had the largest proportion of glycine/arginine (43%), while sample G5-12-18 had the smallest proportion (37%) compared to all samples. All the samples reported more than 30% of glycine/arginine toward the total amino acid content, which agrees with the observations made by Ageeva et al. [3] in their study on lumpfish. The results also agree with the observation that the major amino acid in gelatine is glycine [15,16], which constitutes up to 30% of the collagen matrix [18]. The hydroxyproline content I used to determine the amount of gelatine [25] in the freeze-dried material. Samples G5-12-24 and G5-12-18 had the lowest hydroxyproline content (<5%) and a corresponding lower proportion of gelatine (<60%). Samples G10-18-12 and G15-18-12 presented a higher hydroxyproline content (<7%) and corresponding higher purity of gelatine (>80%). The amino acid composition in cold-water fish gelatine reported by Haug and Draget [26] corresponds with the results of this study, with the exception of the contents of alanine and tyrosine. The proportion of alanine was lower, while tyrosine was higher in our samples. This may indicate the presence of other types of proteins than gelatine in the samples. 

### 2.5. Total Protein and Gelatine Content in the Extracted Gelatine Samples

The total protein content (%) and gelatine content (%) on a dry weight basis are presented in Figure 2. The results show a high content of protein in all the samples. Sample G5-24-24 (94.8%) had the highest total protein content, while sample G10-12-24 (87.4%) had the lowest. Neither the extraction temperature, time, nor ratio showed a significant effect on the total protein content. Gelatine content shows the purity of the extracted freeze-dried samples. Sample G15-18-12 (83 + 1.21%) had the highest gelatine content and thereby the highest purity, while the lowest was found for sample G5-12-18 (57.3 + 3.18%). In this study, the results show that all the samples have a high protein content, while the gelatine content of the samples varies to a greater extent. The results for gelatine content and protein content have a low correlation (r = 0.046), which may be due to the small variability between the samples. Samples G5-12-24, G5-12-18, G5-18-24, and G15-24-24 had a high proportion of protein (>90%) and a lower proportion of gelatine (<60%). A high protein content combined with a lower gelatine content can indicate non-collagenous proteins in the sample [25]. Samples G10-24-18, G10-24-12, G15-12-18, and G15-18-18 reported the highest purity of gelatine (>80%). This indicates that the ratios of 1:10 and above (1:15), an extraction period of 24 h, and an extraction temperature of 18 °C resulted in a maximum level of purity among the tested combinations. An extraction period of 18 h and acid to skin ratio of 1:10 resulted in a higher gelatine content (>76%), while an extraction period of 12 h and acid to skin ratio of 1:5 produced the lowest (<67%). The 1:10 skin to acid ratio has been previously observed by many researchers as an efficient proportion in different cold-water fishes for gelatine extraction [19]. The positive effect of a higher temperature and longer extraction time on gelatine yield was previously demonstrated by Kołodziejska et al. [27] on salmon skin.

### 2.6. Molecular-Weight Distribution

SDS-PAGE was performed on four samples with varying gelatine contents (G10-12-18, G10-18-12, G10-24-24, G10-24-18, and G15-18-12). All the samples show bands in the high-molecular-weight range (Figure 3), approximately between 250 kDa and 100–150 kDa. According to Gerry et al. [28], gelatine from salmon generally contains different molecular-weight protein molecules in the ranges of 70–125 kDa (α-chain), 125–230 kDa (β-chain), and 230–340 (γ-chain). In the present study, the gel image indicates the presence of β and α-chains in the samples, confirming that the source of the extracted gelatine is type-1 collagen [29].

### 2.7. Dry Matter and Ash Content

Five samples with varying gelatine contents (G10-12-18, G10-18-12, G10-24-24, G10-24-18, and G15-18-12) were subjected to dry matter and ash content analyses. The ash content was low (Table 4) (0.40–0.65%) and the dry matter content was high, with a moisture content in the range of 8.77–10.44%. Guerrero et al. [30] observed that fish gelatine in general contained moisture and ash contents of 8.37 ± 0.35% and 1.06 ± 0.11%, respectively, supporting the observations made in this study. According to the standards recommended for food-grade gelatine by the Gelatine Manufacturers of Europe [31], moisture and ash contents should be limited to ≤15% and ≤2%, respectively. The result from the present study indicates that the gelatine extracted from lumpfish skin meets the standard.

### 2.8. Rheological Properties

Three gelatine samples with a gelatine content around 90% (G10-18-12, G10-24-24, and G15-18-12) were selected for rheological analysis. Rheological analysis is based on measurements of melting and gelling temperatures, viscosity, storage modulus, and a comparison of viscosity and shear rate. The result shows only a small variation in the melting (14.2–15.4 °C) and gelling (6–7 °C) temperatures among the samples (Table 5). Storage modulus, or gel strength, is a measure of how much energy must be supplied to deform the sample [32]. All the samples presented a storage modulus between 2869 and 3801 pascals. The viscosity at a shear rate of 0.1–1 s ranged between 15 to 37 Pa s, and at a shear rate of 10 s^−1^ was between 11 to 24 Pa s. 

Fish gelatine, particularly that sourced from cold-water fish resources, has been observed to possess low gelling and melting temperatures, storage modulus, and viscosity properties compared to mammalian gelatine [33]. The melting and gelling temperatures of lumpfish gelatine reported in this study were in the same range as those observed for salmon gelatine [32]. The storage modulus values were also in the range of 2000–3000 Pas as previously reported for fish gelatine prepared using alkaline pre-treatment and acetic acid hydrolysis [34]. The gel strength is observed to be mainly dependent on the molecular-weight distribution of gelatine samples [35]. Studies have shown that gelatine fragments with a lower molecular weight are negatively correlated with gel strength, while α-chains, β-chains, and molecules with a higher molecular weight are positively correlated [13]. The viscosity is also found to be positively correlated with higher-molecular-weight molecules [36]. This study also observed shear thinning behavior, where an increase in the shear rate presented a decrease in the viscosity of the gelatine samples, as previously observed in the case of salmon gelatine [37]. This was due to the constant alignment of the highly anisotropic chains in the course of the shear rate [38] and also in connection with the alterations in gel dispersion, the shape of the molecules, and intra-molecular bonds according to the shear force [39]. The low viscosity and gelling temperatures of cold-water fish gelatine, such as lumpfish, were observed to be suitable for food-coating applications, as the industry prefers lower setting temperatures [37]. This property is also reported to be advantageous in comparison to mammalian gelatine in bio-fabrication techniques, where low gelling and viscosity properties assist in therapeutic treatments and tissue engineering [40]. These properties make them suitable for applications where miniature structure constructions with precise geometric alignments are prepared using techniques such as micropatterning, microfluidics, and electrospinning [41,42,43]. Other than edible food coatings, cold-water marine collagen can also be used for polymer-based scaffolds and bio-ink for tissue engineering-based 3D printing [44,45,46]. C-water marine collagen is also superior to mammalian collagen where a lower working temperature and higher concentrations is required as these conditions would lead to the clotting of the mammalian collagen, making it ineffective [47]. 

## 3. Materials and Methods

### 3.1. Preparation of Raw Material 

Frozen lumpfish carcasses after the removal of roe and viscera were received from Royal Greenland AS and Biopol/University of Akureyri. The carcasses were stored at −40 °C until further processing. The carcasses were thawed at 4 °C for approximately 18 h and carefully dissected by hand and separated as skin, meat, and frames, and the individual yield (% wt. of total carcass) was determined. The skin was cut into pieces of roughly 1 cm^2^, rinsed with water, weighed, and frozen at −20 °C until further processing. Prior to the extraction process, the lumpfish skin was thawed in cold water (4 °C) for approximately one hour.

### 3.2. Extraction of Gelatine

A modified version of the method suggested by Eysturskarð et al. [48] was used for extracting gelatine from thawed lumpfish skin. The extraction process was divided into three stages: pre-treatment, acid extraction, and freeze-drying. Figure 4 shows the general flow of the pre-treatment and extraction processes, and Table 6 indicates different combinations of extraction parameters and the respective sample codes. 

#### 3.2.1. Pre-Treatment Process

Thawed lumpfish skin was treated with 0.1 M of sodium hydroxide (NaOH) (VWR International Ltd., Oslo, Norway) for removing the non-collagenous protein [49] at a ratio of 1:10 for 24 h with gentle shaking (100 rpm) in an incubator (New Brunswick™ Innova^®^ 42/42R Benchtop Incubator Shakers, Stackable, Eppendorf, Hamburg, Germany) at 5 °C. After the alkaline pre-treatment, the skin was filtered through a cheese cloth and washed repeatedly in cold water until the pH was neutral. The de-proteinized and washed lumpfish skin was then subjected to de-mineralization in order to remove calcium from the skin using 0.1 M of hydrochloric acid (HCl) (VWR International Ltd., Oslo, Norway) with similar ratio and treatment conditions as those of the alkaline treatment. The de-mineralized skin was further filtered and washed with chilled water to achieve a neutral pH. 

#### 3.2.2. Acid Extraction

Gelatine was extracted from the pretreated lumpfish skin with 0.1 M of acetic acid (VWR International Ltd., Oslo, Norway) with three different extraction ratios, extraction times, and extraction temperatures, as indicated in Table 6, using an incubator (New Brunswick™ Innova^®^ 42/42R Benchtop Incubator Shakers, Stackable, Eppendorf) shaken at 100 rpm. 

#### 3.2.3. Freeze-Drying

After the acid treatment, the remaining solid particles were filtered out using a cheese cloth, and the remaining liquid portion was transferred to round-bottom flasks and frozen by immersion in liquid nitrogen. The frozen samples were then loaded to a Christ Alpha 1–4 LD freeze-dryer (Martin Christ Gefriertrocknungsanlagen GmbH, Osterode, Germany). The freeze-dried gelatine samples were weighed and stored at room temperature in zip-lock pouches until further analysis. 

### 3.3. Analysis

The yield and proximate composition (water, protein, lipid, and ash contents) of the different fractions (skin, bone, and muscle) of the raw lumpfish carcass were determined. For the lumpfish skin, the total amino acid composition, including content of hydroxyproline, was determined in order to estimate the quality of the raw material for gelatine extraction. For the extracted gelatine, the yield, total amino acid composition, hydroxyproline content, and total protein content were analyzed. On the basis of the gelatine yield, five samples were selected for the further analysis of the molecular-weight distribution (SDS-PAGE), dry matter, ash content, and rheological properties. All analyses were performed with a minimum of three replicates. 

#### 3.3.1. Proximate Composition and Total Amino Acid Composition of the Lumpfish Fractions 

The moisture content was analyzed by drying the carcass components (skin, meat, and frame) (2.0 g) at 104 °C for 20 h [50]. The ash content was determined by combusting the pre-weighed samples in a muffle furnace at 540 °C for 16 h [51]. Crude protein was determined using the Kjeldahl titration method [52] and the lipid content was determined by the Bligh and Dyer method [53]. The determination of the total amino acid composition of the freeze-dried skin sample was performed as described by Blackburn [54], using a reversed-phase Ultra High-Performance Liquid Chromatography (RP-HPLC) (Dionex UltiMate^®^ 3000 UHPLC+ Focused, Dionex UltiMate^®^ 3000 Autosampler, Dionex RF Fluorescence Detector, Thermo Scientific, Waltham, MA, USA) and NovaPak column (Nova-Pak C18 4 μm, 3.9·150 mm, Waters tech. Ltd, Wexford, Ireland). The method cannot detect proline, hydroxyproline, hydroxylysine, or cysteine. Glycine and arginine are determined together as their peaks merge.

#### 3.3.2. Extraction Yield, Total Amino Acid Composition, and Hydroxyproline and Gelatine Content Analyses of the Freeze-Dried Gelatine Samples

The total extraction yield was calculated using the weight of the freeze-dried extract against the dry weight of the lumpfish skin and expressed as a percentage. The total amino acid composition of the freeze-dried gelatine samples was determined as described in Section 3.3.1. The hydroxyproline content of the freeze-dried gelatine samples was determined according to the spectrophotometric method described by Leach [55], which is a modified method by Neuman and Logan [56]. The OD of the samples was measured at 555 nm using a spectrophotometer (Ultrospec 2000 UV/Visible spectrophotometer, Pharmacia Biotech, Uppsala, Sweden). To convert the amount of hydroxyproline to determine the gelatine content, a conversion factor of 11.7 was used as it represents the hydroxyproline (g/100 g) content in lumpfish gelatine.

#### 3.3.3. Protein Analysis

The protein content of the freeze-dried samples was determined using the C/N/S elemental analysis method. A C/N elemental analyzer (Elementar Combustion System CHNS-O, ECS 4010, Langenselbold, Germany) was used to determine the total nitrogen content in the experimental units. A protein conversion factor of 5.55 was used with the assumption that the gelatine was completely pure [57].

#### 3.3.4. Molecular-Weight Distribution

The molecular-weight distribution of the freeze-dried gelatine samples was determined using sodium dodecyl sulphate polyacrylamide gel electrophoresis (SDS–PAGE), according to He [58], using a C.B.S. Dual Cool System (DCX-700). An Invitrogen™ NuPAGE™ 4 to 12%, Bis-Tris, 1.0 mm, 12-well SDS gel was used with a high-molecular-weight standard (Thermo Scientific™ PageRuler Unstained Broad Range Protein Ladder). The apparatus was connected to a PS300B 300 volt power supply (AA Hoefer, Holliston, MA, USA). After electrophoresis, the gel was removed from the cassette and rinsed briefly with distilled water. The gels were stained using an InstantBlue^®^ Coomassie Protein Stain (ISB1L). The analyses were carried out according to the instructions from the manufacturer. 

### 3.4. Rheological Properties

A Kinexus rheometer (Kinexus ultra+, Westborough, MA, USA), which measures the amount of deformation or strain the material reaches when it is subjected to oscillating movements with small loads, was used for the measurement. The purpose of the analysis was to map the melting and gelling temperatures, viscosity, and gel strength of the five selected freeze-dried gelatine samples. A plate geometry (d = 40 mm, 4°) with a 0.15 mm gap was used with a frequency of 1 Hz, constant shear strain of 0.50%, and a temperature gradient of 1 °C/min for both cooling and heating. The freeze-dried gelatine was dissolved in distilled water to a concentration of 10% (*w*/*v*), where 2.45 mL of the sample was added to the plate geometry. The samples were sealed with low-viscosity silicone oil (200/10cS fluid, Dow Corning Corporation, London, UK) to prevent evaporation. The start and end temperatures were set to 20 °C, while the curing temperature was set to 4 °C. The change in temperature in the process was 20 °C → 4 °C → 20 °C. Viscosity vs shear rate was measured after reaching 20 °C for the second time.

### 3.5. Statistical Analysis

The results are presented as mean ± standard deviation, and were treated in the statistics program IBM SPSS Statistics, version 29 (IBM, New York, NY, USA). A normal distribution of all the data was assumed. The general linear model (GLM) using the Tukey’s post hoc test was used to test the statistical differences between groups and within groups. Pearson’s correlation coefficient was used to test the correlation.

## 4. Conclusions

The study explored acid hydrolysis as a tool for utilizing largely underutilized Atlantic lumpfish *(Cyclopterus lumpus)* carcass skin as a source of marine gelatine. The study demonstrated that a higher extraction time (18–24 h) and extraction temperature (18–24 °C) produced the maximum extraction yield and gelatine purity (>80%), while the skin:acid ratio had a negligible effect. The molecular weight and rheological studies are consistent with the result that the source of the extracted gelatine is type-I collagen, with typical cold-water fish gelatine properties, like a high molecular weight, low melting and gelling properties, and lower gel strength and viscosity characteristics, suitable for specific food-grade applications. The method could be adopted by the Atlantic lumpfish fishing and roe industry for effectively utilizing the side streams generated with a sustainable objective.

## Figures and Tables

**Figure 1 marinedrugs-22-00169-f001:**
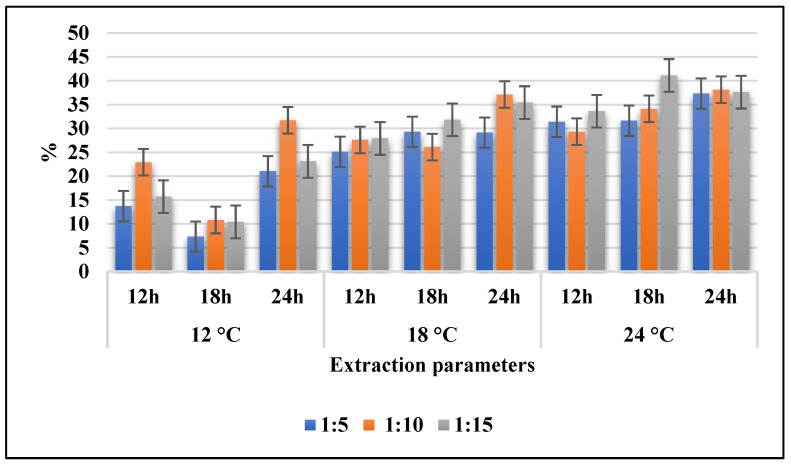
Total extraction yield (%) in dry weight.

**Figure 2 marinedrugs-22-00169-f002:**
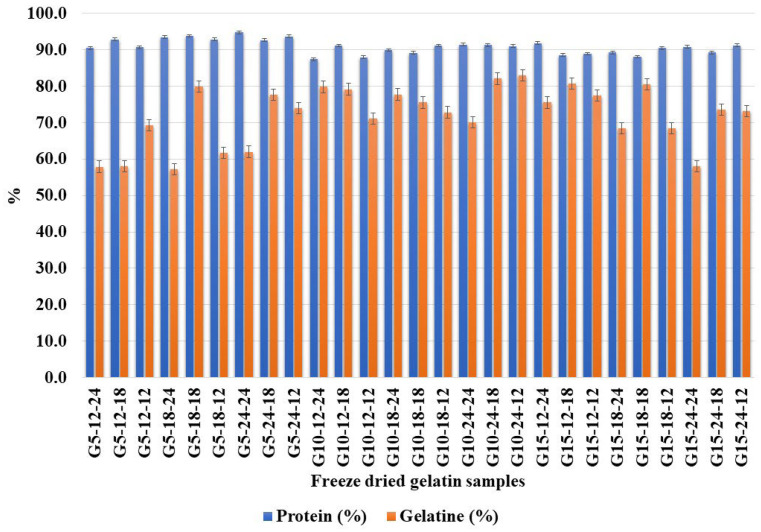
Protein and gelatine contents in the extracted samples.

**Figure 3 marinedrugs-22-00169-f003:**
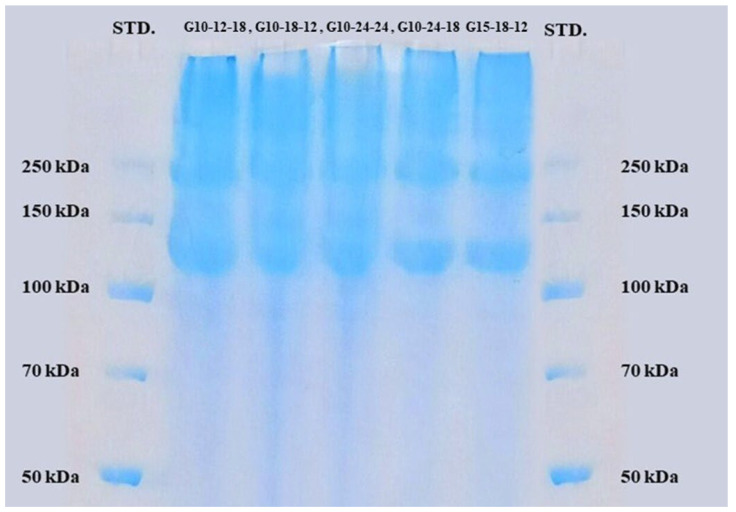
SDS-PAGE image of the selected samples.

**Figure 4 marinedrugs-22-00169-f004:**
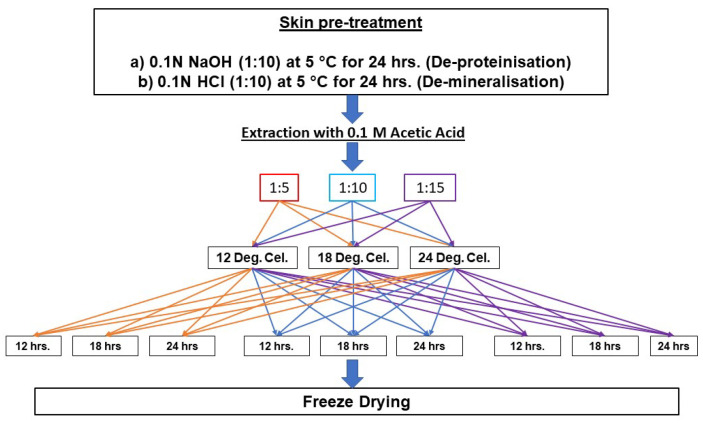
Complete flow chart of the experiment.

**Table 1 marinedrugs-22-00169-t001:** Proximate and yield composition of lumpfish carcass.

Sample	Moisture %	Protein %	Lipid (%)	Ash (%)	Yield (%)
Skin	86 ± 1.00	11.66 ± 0.65	0.62 ± 0.11	1.38 ± 0.27	41.4 ± 0.37
Meat	82 ± 1.4	7.25 ± 0.02	10.72 ± 0.35	1.01 ± 0.02	24 ± 0.42
Frames	86 ± 0.41	7.53 ± 0.32	3.42 ± 0.65	1.34 ± 0.07	34.6 ± 0.53

The results are presented as mean ± standard deviation. *N* = 3.

**Table 2 marinedrugs-22-00169-t002:** Total amino acid content of lumpfish skin.

Amino Acid	mg/g Freeze-Dried Skin
Alanine	50.42 ± 10.51
Glycine/arginine	257.99 ± 52.02
Aspartic acid	44.76 ± 8.99
Glutamic acid	56.16 ± 11.44
Asparagine	0.02 ± 0.00
Histidine	4.6 ± 3.79
Serin	41.33 ± 8.97
Glutamine	0.45 ± 0.25
Threonine	24.43 ± 4.81
Tyrosine	3.6 ± 2.08
α-Aminobutyric acid	0.59 ± 0.13
Methionine	12.03 ± 2.51
Valin	15.98 ± 3.48
Phenylalanin	15.67 ± 3.52
Isoleucine	9.37 ± 1.86
Leucin	26.51 ± 5.05
Lysin	25.38 ± 5.25
Hydroxyproline	40 ± 0.09
Total	631.27 ± 119.96

The results are presented as mean ± standard deviation. *n* = 3.

**Table 3 marinedrugs-22-00169-t003:** Total amino acid profile of lumpfish gelatine samples.

Amino Acids (mg/g)	G5-12-24	G5-12-18	G5-12-12	G5-18-24	G5-18-18	G5-18-12	G5-24-24	G5-24-18	G5-24-12	G10-12-24	G10-12-18	G10-12-12	G10-18-24	G10-18-18	G10-18-12	G10-24-24	G10-24-18	G10-24-12	G15-12-24	G15-12-18	G15-12-12	G15-18-24	G15-18-18	G15-18-12	G15-24-24	G15-24-18	G15-24-12
Aspartic acid	51 ± 6	35 ± 9	49 ± 3	45 ± 1.5	44 ± 5	44 ± 5	45 ± 4	49 ± 3	44 ± 4	42 ± 9	42 ± 4	47 ± 10	44 ± 3	41 ± 3	41 ± 4	48 ± 6	44 ± 6	42 ± 6	44 ± 2	39 ± 8	53 ± 8	39 ± 15	37 ± 4	41 ± 4	44 ± 6	51 ± 6	42 ± 5
Glutamic acid	61 ± 9	73 ± 11.2	64 ± 5	59 ± 2	57 ± 7	58.5 ± 7	59 ± 6	63 ± 6	59 ± 5	52 ± 13	55 ± 4	61 ± 13	61.3 ± 4	54 ± 4	54 ± 7	63 ± 6	58 ± 8	55 ± 8	55 ± 17	53.6 ± 2	69 ± 11	54.54 ± 21	51 ± 10	55 ± 5	57 ± 8	67 ± 8	56 ± 7
Histidine	5.4 ± 1	2 ± 1	3 ± 1.4	7.4 ± 0.4	4 ± 1	4 ± 1	5 ± 3	3 ± 1	2 ± 0.9	1 ± 0.3	4 ± 0.4	4 ± 2	1.08 ± 0.05	4 ± 2	5 ± 0.1	3 ± 2	6 ± 2	5 ± 2	3 ± 2	4 ± 1	5 ± 3	4 ± 2	3 ± 1	5 ± 2	5 ± 3	5 ± 3	4 ± 2
Serin	50 ± 10	35.4 ± 10	55 ± 3	53 ± 2	50 ± 7	47 ± 6	52 ± 6	53 ± 5	49 ± 5	44 ± 12	44 ± 5	53 ± 12	44 ± 4	46 ± 5	44 ± 6	53 ± 6	51 ± 7	48 ± 8	47 ± 15	40 ± 10	65 ± 9	40 ± 16	36 ± 8	45 ± 6	50 ± 8	57 ± 8	44 ± 6
Glycine + Arginine	380 ± 41	230 ± 61.4	361 ± 23	323 ± 12	310 ± 40	299 ± 36	313 ± 34	355 ± 23	326 ± 34	285 ± 61	274 ± 27	346 ± 83	289 ± 31	290 ± 25	288 ± 33	351 ± 36	306 ± 43	292 ± 45	319 ± 103	254 ± 48	394 ± 61	264 ± 105	245 ± 26	291 ± 26	303 ± 44	374 ± 45	284 ± 41
Threonine	21 ± 3	20 ± 4	21 ± 1.5	25 ± 1	24 ± 3	16 ± 3	24.4 ± 3	21 ± 3	19 ± 2	21 ± 5	18 ± 6	22 ± 2	18 ± 6	22 ± 2	20 ± 4	21 ± 3	24 ± 3	23 ± 3	20 ± 4	18 ± 1	23 ± 3	20 ± 7	18 ± 3	21 ± 3	24 ± 4	22 ± 3	20 ± 8
Tyrosine	162 ± 16	103 ± 26	155 ± 12	140 ± 5	136 ± 15	136 ± 18	137 ± 14	149 ± 7	136 ± 11	125 ± 26	123 ± 10	146 ± 36	131 ± 9	128 ± 12	129 ± 15	150 ± 18	134 ± 16	127 ± 18	138 ± 44	113 ± 23	167 ± 28	116 ± 46	110 ± 11	128 ± 12	131 ± 19	158 ± 18	126 ± 16
Alanine	0.10 ± 0.1	0.54 ± 0.04	0.02 ± 0.002	1.2 ± 0.3	0.44 ± 0.1	0.51 ± 0.07	0.5 ± 0.08	0.03 ± 0.03	0.04 ± 0.02	0.01 ± 0.01	2 ± 0.2	0.03 ± 0.02	0.11 ± 0.03	0.50 ± 0.07	1 ± 0.08	0.03 ± 0.01	0.5 ± 0.07	0.20 ± 0.21	0.05 ± 0.01	0.54 ± 0.06	0.66 ± 0.23	0.98 ± 0.80	0.00	1 ± 0.05	1 ± 0.7	0.54 ± 0.08	1 ± 0.08
Methionine	12.5 ± 0.003	8.5 ± 0.05	12.2 ± 0.005	11 ± 0.1	12 ± 0.1	11 ± 0.2	12 ± 0.1	13 ± 0.01	11.5 ± 0.09	11 ± 0.03	9 ± 0.1	13 ± 0.1	10 ± 0.03	10 ± 1	10 ± 0.1	13 ± 0.8	11 ± 0.03	11 ± 0.02	11 ± 0.05	8 ± 0.05	14 ± 0.9	8 ± 0.02	9 ± 0.02	10 ± 0.1	11 ± 0.03	13 ± 0.01	10 ± 0.03
Valin	17 ± 2	12.3 ± 3	18 ± 1	153 ± 0.6	15 ± 0.9	14 ± 2	15 ± 0.8	18 ± 1	16 ± 1	16 ± 3	14 ± 1	17 ± 3.5	14 ± 0.4	13 ± 2	13 ± 2	18 ± 2	14 ± 0.5	14 ± 2	16 ± 3	12 ± 2	19 ± 2	12 ± 4	13 ± 1	13 ± 2	15 ± 1	19 ± 2	14 ± 2
Phenylalanin	19 ± 3	11 ± 4	17 ± 1.2	16.4 ± 0.5	15 ± 1.5	15 ± 2	16 ± 1	17 ± 1.5	15 ± 1.5	14 ± 4	16 ± 1	16.5 ± 4	14 ± 0.6	14 ± 2	15 ± 2	17 ± 3	16 ± 2	15 ± 2	15 ± 5	13 ± 3	18 ± 3	13 ± 5	12 ± 2	15 ± 2	16 ± 2	18 ± 2	15 ± 2
Isoleucine	8 ± 2	5.4 ± 3.5	8.4 ± 1.2	7 ± 0.6	7 ± 2.3	6 ± 1	7 ± 2	9 ± 1	8 ± 1.5	7 ± 3	6 ± 1	8 ± 4	6 ± 1	6 ± 1	6 ± 2	9 ± 1	6.4 ± 2	6 ± 2	7 ± 1	5 ± 1.6	9 ± 3	5 ± 1.4	5 ± 1	6 ± 2	7 ± 2	9 ± 2	6 ± 3
Leucin	22 ± 2	12 ± 1.4	20 ± 0.6	18 ± 0.7	18 ± 1	15 ± 1	19 ± 1	21 ± 0.6	19 ± 0.6	17 ± 2	16 ± 1	19 ± 2	14 ± 0.3	14 ± 1	14 ± 1	21 ± 1	19 ± 1	18 ± 1	18 ± 3	13 ± 1	22 ± 2	12 ± 2	14 ± 1	14 ± 0.5	19 ± 1	22 ± 1	15 ± 1
Lysin	29.5 ± 2.4	19 ± 3	28 ± 1.3	26 ± 1	25 ± 2.5	25.5 ± 2	25 ± 2	27.5 ± 1	25 ± 2	23 ± 4	23 ± 1.5	27 ± 4.5	24 ± 1	24 ± 1	24 ± 2	27 ± 2	24 ± 3	23 ± 3	25 ± 6	21 ± 3	30 ± 3	21 ± 5	23 ± 1	24 ± 1	25 ± 3	29 ± 3	24 ± 2
Hydroxyproline	49 ± 3	49 ± 5	53 ± 2	68 ± 1	66 ± 3.3	60 ± 3.4	65 ± 3	58.5 ± 2	50 ± 2	50 ± 5	68 ± 3	66 ± 6	68 ± 2	65 ± 2	70 ± 3	69 ± 3	69 ± 4	63 ± 4	59 ± 8	53 ± 5	63 ± 5	61 ± 8	62 ± 6	71 ± 2	66 ± 4	59 ± 3	62 ± 3
Total Amino Acid	887 ± 94	615 ± 155.5	865 ± 54.3	815 ± 29.5	783 ± 92	753 ± 83	794 ± 80	856 ± 54	777 ± 69	705 ± 146	714 ± 60	846 ± 179	738 ± 61	731 ± 58	733 ± 76	859 ± 84	782 ± 97	741 ± 104	776 ± 228	646 ± 119	946 ± 141	670 ± 242	638 ± 72	741 ± 67	773 ± 103	902 ± 104	725 ± 96

**Table 4 marinedrugs-22-00169-t004:** Dry matter and ash contents in lumpfish gelatine.

Sample	Dry Matter (%)	Ash (%)
G10-12-18	89.53 ± 0.26	0.63 ± 0.72
G10-18-12	91.23 ± 0.39	0.65 ± 0.93
G10-24-24	89.04 ± 0.41	0.55 ± 0.84
G10-24-18	90.29 ± 0.52	0.40 ± 0.95
G15-18-12	89.98 ± 0.31	0.58 ± 0.64

The results are presented as mean ± standard deviation. *n* = 3.

**Table 5 marinedrugs-22-00169-t005:** Rheological properties of lumpfish gelatine.

Gelatine Sample	Gelling Temperature (°C)	Melting Temperature (°C)	Storage Modulus (G′) (2 h, 4 °C) in Pa	Viscosity (Pa s) at 0.1 s^−1^	Viscosity (Pa s) at 10 s^−1^
G10-18-12	6.6 ± 1.2	14.5 ± 1.81	3801 ± 2.8	0.28 ± 0.07	0.17 ± 0.02
G10-24-24	6 ± 1.61	14.2 ± 1.39	2869 ± 3.4	0.15 ± 0.02	0.11 ± 0.05
G15-18-12	7 ± 0.95	15.4 ± 1.26	3147 ± 2.45	0.37 ± 0.01	0.24 ± 0.01

The results are presented as mean ± standard deviation. *n* = 3.

**Table 6 marinedrugs-22-00169-t006:** Different sample codes and extraction parameters.

Sample Code	Skin:Acid Ratio	Reaction Time (h.)	Reaction Temperature (°C)
G5-12-24	1:5	12	24
G5-12-18	1:5	12	18
G5-12-12	1:5	12	12
G5-18-24	1:5	18	24
G5-18-18	1:5	18	18
G5-18-12	1:5	18	12
G5-24-24	1:5	24	24
G5-24-18	1:5	24	18
G5-24-12	1:5	24	12
G10-12-24	1:10	12	24
G10-12-18	1:10	12	18
G10-12-12	1:10	12	12
G10-18-24	1:10	18	24
G10-18-18	1:10	18	18
G10-18-12	1:10	18	12
G10-24-24	1:10	24	24
G10-24-18	1:10	24	18
G10-24-12	1:10	24	12
G15-12-24	1:15	12	24
G15-12-18	1:15	12	18
G15-12-12	1:15	12	12
G15-18-24	1:15	18	24
G15-18-18	1:15	18	18
G15-18-12	1:15	18	12
G15-24-24	1:15	24	24
G15-24-18	1:15	24	18
G15-24-12	1:15	24	12

## Data Availability

The original contributions presented in the study are included in the article. Further inquiries can be directed to the corresponding author.

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
