# Peer review of "Utilization of Lumpfish (Cyclopterus lumpus) Skin as a Source for Gelatine Extraction Using Acid Hydrolysis"

_marinedrugs, 2024, doi:10.3390/md22040169_

Round 1

Reviewer 1 Report

Comments and Suggestions for Authors

The manuscript is devoted to an important topic related to the release of gelatin from fish waste products. This allows us to solve two problems: obtaining a valuable biomedical food product and the disposal of biological waste. The article contains a large amount of material, which allows for a full-fledged analysis of the results obtained and suggests the most optimal conditions. However, there are a number of disadvantages and issues:

1. The most important point is related to the terminology used. I believe that gelatin was not isolated in this study. and collagen. Gelatin is a product of collagen denaturation. Denaturation of fish collagen should occur at temperatures above the maximum temperature of the habitat. The water temperature, for example, in the Barents Sea can reach 25 ° C, therefore, collagen denaturation in the skin should begin after this temperature. The presented studies were mainly conducted at a lower temperature, therefore, we are talking about collagen.

Please explain the attribution of the isolated protein specifically to gelatin. Have you conducted any studies confirming this?

In the conclusions, “The molecular weight and rheological studies established that the extracted gelatine was food grade type-I collagen with typical cold water fish gelatine properties like high molecular weight, low melting and gelling properties, lower gel strength and viscosity characteristics suitable for specific food grade applications.”

So what is it all about Is the speech in the manuscript? About gelatin or collagen? It is necessary to clarify the text.

2. In Table 4 and its description, the change in the composition of collagen in some cases is associated with the extraction of other proteins. Why then was the pretreatment of the skin of the pinagore NaOH and HCl carried out? What was highlighted at this stage? And how was it monitored?

3. I would like to see clearer prospects for the use of the resulting gelatin, and not only as an abstract element of sustainable development. Then such a voluminous array of research will become clear.

 I believe that the article can be published with minor changes.

Author Response

The answer to the reviewers has been uploaded as a PDF

Reviewer 2 Report

Comments and Suggestions for Authors

The research topic is interesting for the readers, as well as the praxis, as the paper focuses on using by-product fish skin to extract gelatins. The experimental design of gelatin extraction used GLM with the Tukey post-hoc test; 3 processing parameters were tested. A vast number of analyses have been made to characterize prepared gelatins. Prepared gelatins are of high quality regarding low ash content. The gelling and setting points of gelatins are low, but this is typical for fish gelatins.

Nevertheless, some questions need to be explained, and some misunderstandings need to be clarified/revised:

1. Check the overall formal style of the paper, e.g., page number.

2. The font of Figure 1 should be the same as in the text body (Palatino Linotype).

3. Chapter 2.3.2., last sentence (determining gelatin content from HYP): reference could be added.

4. Chapter 3.2; amino acid composition of raw material: In the first sentence, authors say that glycine represents approx. 33 % of the total amino acids in the sample. On the other hand, in Table 3, the content of glycine+arginine is approx. 26%. Clarify/revise.

5. Chapter 3.4; sentence in lines 245-247. I do not understand the explanation regarding the proportion of glycine/proline in compared samples. G5-12-24 (43 %), G5-12-18 (37%). Looking at Table 4, in the G5-12-24 sample, the amount of glycine is 380 mg/g and arginine 41 mg/g; in the G5-12-18 sample, the amount of glycine is 230 mg/g and arginine 61.4 mg/g. How do these figures correlate with the proportion figures?

6. Chapter 3.6: Figure 4 is missing.

7. Chapter 3.8: In the text, use the capital “P” in “pascal”.

Author Response

the answer to the reviewer comments have been uploaded as a PDF file

Reviewer 3 Report

Comments and Suggestions for Authors

Comments

1. The main note throughout the article is that the authors in the test use the names of the substance “collagen” and gelatin,” without identifying how collagen and gelatin differ. It is known that in many respects, molecular weight, amino acid composition and some others, these substances isolated from the same material do not differ noticeably. But these are different substances! Collagen is a thermally unstable polymer: at temperatures above 30-400C, its denaturation begins with the formation of gelatin. The difference between collagen and gelatin is that the collagen molecule is a left-handed helix of three α-chains from amino acid residues of known amino acids around a common axis. In aqueous solutions, such a macromolecule is further stabilized due to hydrogen bonds with water. The gelatin molecule is a denatured helix with broken bonds of individual α-chains and in aqueous solutions forms aggregates between the intermolecular free parts of the α-chains in solution. Due to this, the polypeptide chains of gelatin in dilute aqueous solutions form compact coils that are partially permeable to the solvent. At the same time, such characteristics as viscosity, glass transition temperature, etc., change significantly in comparison with collagen. This should always be taken into account when carrying out developments. Taking into account the above, in my opinion, the phrase from the conclusions looks absurd:

  “The molecular weight and rheological studies established that the extracted gelatine was food grade type-I collagen with typical cold water fish gelatine properties like high molecular weight, low melting and gelling properties, lower gel strength and viscosity characteristics suitable for specific food grade applications”.

The authors should clearly separate the concepts: where they obtain experimental data with gelatin, and where with collagen. Remove phrases such as the one above from the text of the article.

2. In the introduction it is stated that the research was designed to identify the optimal process, taking into account various combinations of extraction, for maximum yield and quality of gelatin. The conclusions indicate only the optimal temperature and time for extraction with acetic acid solution. Where are all the other process parameters? Conditions for pre-treatment with alkaline and hydrochloric acid solution, extraction coefficient, conditions for extracting collagen and skin residues from the mixture, etc.?

The authors should clearly formulate the purpose of the work and, in accordance with the experiment performed, draw conclusions about the work.

3. In clause 2.3.1 there is the phrase “to constant weight at 104 °C for 20 hours.” What if, after 20 hours of drying, the mass of the sample continues to decrease?

It is customary to formulate “up to constant weight at 104 °C.”

4. In paragraph 2.3.2 there is the phrase “The total amino acid composition of lyophilized gelatin samples was determined as described in section 1.3.1.

There is no section 1.3.1 in the text.

5. In clause 2.3.3. indicated: Protein content in lyophilized samples.” Further in the text in paragraph 3.1. “Skin had the highest amount of protein (11.66 ± 0.65%).” Apparently, we are talking about the original non-dehydrated carcass mass?

It should be clarified in the text that when the amount of protein is indicated (11.66 ± 0.65%), this is in relation to the mass of which substrate?

6. The text contains the phrase “Total protein content (%) and gelatin content (%) on a dry weight basis are presented in Figure 3. The results show high protein content in all samples. Sample G5-24-24 (94.8%) had the highest total protein content, and sample G10-12-24 (87.4%) had the lowest.

In Fig. 3, the total protein content in all samples is 100%.

The data in Fig. should be brought into line. 3 with the text of the article.

This material raises two more questions:

1) How correct is it to compare the results of determining total protein in samples by nitrogen content (Kjeldahl method) and collagen by hydroxyproline? First, both methods allow for large detection errors; secondly, the authors chose a conversion factor for the protein content in the samples of 5.55; according to the article the authors refer to, this is for gelatin. And the content of total protein apparently suggests other proteins? Then the coefficient should be different?

All this should be explained in the text of the article and changes should be made to the text.

2) In clause 2.1.1. The authors describe a method of washing with 0.1 M sodium hydroxide to remove non-collagenous protein, and then state in paragraph 3.5 that there is quite a lot of non-collagenous protein in the samples.

Why did it happen? Is the protein washing method ineffective? Is it possible that it is not correct to compare the indicators of protein according to Kjeldahl and collagen according to hydroxyproline?

7. In clause 3.6. there is a link to Fig. 4, but Fig. 4 is not in the text of the article!

Figure 4 should be included in the text. Only after this can one comment on the molecular weight data.

Author Response

The answer to the reviewers comments have been uploaded as a PDF file

Round 2

Reviewer 3 Report

Comments and Suggestions for Authors

1.The "Conclusions" section has an incorrect numbering.

2. The section 3.3.1. again states that the samples were dried for 20 hours, and not to a constant weight.

3. The authors should check the entire numbering of the sections

4. There is no need for a separate section about "Dry matter and ash content analysis of gelatin samples" if all the information from it is contained in other sections.

Author Response

Thank you for pointing this out.

The statement on dry matter determination has been corrected and the separate section on dry matter and ash content has been removed. The numbering of the sections has been revised and corrected.